# Epidemiology and control of maedi-visna virus: Curing the flock

**Andrew W. Illius[1], Karianne Lievaart-Peterson[2], Tom N. McNeilly[3], Nicholas J. Savill[4]***

**1** Institute of Evolutionary Biology, School of Biological Sciences, University of Edinburgh, Edinburgh, Scotland, **2** GD Animal Health, Deventer, The Netherlands, **3** Moredun Research Institute, Pentlands Science Park, Penicuik, Midlothian, Scotland, **4** Institute of Immunology and Infection Research, School of Biological Sciences, University of Edinburgh, Edinburgh, Scotland

* nick.savill@ed.ac.uk

**Data Availability Statement:** All relevant data are within the manuscript and its Supporting Information files.

**Funding:** TNM receives funding from the Scottish Government Rural Affairs, Food and the

## Abstract

Maedi-visna (MV) is a complex lentiviral disease syndrome characterised by long immunological and clinical latencies and chronic progressive inflammatory pathology. Incurable at the individual level, it is widespread in most sheep-keeping countries, and is a cause of lost production and poor animal welfare. Culling seropositive animals is the main means of control, but it might be possible to manage virus transmission effectively if its epidemiology was better quantified. We derive a mathematical epidemiological model of the temporal distributions of seroconversion probabilities and estimate susceptibility, transmission rate and latencies in three serological datasets. We demonstrate the existence of epidemiological latency, which has not explicitly been recognised in the SRLV literaure. This time delay between infection and infectiousness apparently exceeds the delay between infection and seroconversion. Poor body condition was associated with more rapid seroconversion, but not with a higher probability of infection. We estimate transmission rates amongst housed sheep to be at about 1,000 times faster than when sheep were at grass, when transmission was negligible. Maternal transmission has only a small role in transmission, because lambs from infected ewes have a low probability of being infected directly by them, and only a small proportion of lambs need be retained to maintain flock size. Our results show that MV is overwhelmingly a disease of housing, where sheep are kept in close proximity. Prevalence of MV is likely to double each year from an initial low incidence in housed flocks penned in typically-sized groups of sheep (c. 50) for even a few days per year. Ewes kept entirely at grass are unlikely to experience transmission frequently enough for MV to persist, and pre-existing infection should die out as older ewes are replaced, thereby essentially curing the flock.

## Introduction

Maedi-Visna (MV) is an insidious, lifelong and eventually fatal disease syndrome in sheep and occurs in most sheep-keeping areas worldwide. The most common symptoms, chronic respiratory disease and indurative mastitis, only become evident some years after infection, and

Environment (RAFE) Strategic Research Portfolio 2016-2021. The funders had no role in study design, data collection and analysis, decision to publish, or preparation of the manuscript.

**Competing interests:** The authors have declared that no competing interests exist.

cause lost production and excess resource use [1]. MV, called ovine progressive pneumonia (OPP) in North America, and its equivalent in goats (caprine arthritis encephalitis, CAE) are caused by the small ruminant lentiviruses (SRLV, family retroviridae), recognised as a heterogeneous group of viruses that infect sheep, goats and wild ruminants with evidence of cross-species infection [2, 3]. Breed differences in susceptibility and the genetics of resistance to SRLV are recognised [4–6]. The slow appearance of disease symptoms which characterises lentivirus infections has important implications for their epidemiology and control at both flock and regional levels, as well as for health and welfare. There is no vaccine nor a cure for the treatment of infected individuals and the methods for achieving an accredited virus-free flock are costly: testing, culling or separating seropositive animals and restocking [7]. We show here that curing the flock of SRLV should be possible by exploiting its transmission characteristics.

Lentiviruses target the immune system by infecting primarily the monocyte/macrophage lineage, but SRLV do not target T-cells as do immune deficiency lentiviruses such as HIV. They persist by latency and immune evasion, including by antigenic drift and mutations of neutralisation epitopes [8]. Sheep infected by SRLV show an initial viraemic phase, but subsequently virus is mostly cell-associated with minimal free virus circulating peripherally [1]. Circulating infected monocytes infiltrate the interstitial spaces of target organs such as lung, mammary gland, or the synovial tissue of joints, carrying proviral DNA integrated into the host cell genome and hence invisible to the immune system. Seroconversion is delayed, typically by several months [9]. Virus replication commences following maturation of monocytes into macrophages, and the ensuing immune response slowly and progressively causes chronic inflammatory lesions and gross pathology [10, 11]. These take years to present clinical symptoms, yet infected sheep are apparently a persistent source of virus for transmission [12].

Transmission of SRLV is now recognised as being primarily horizontal, through inhalation of respiratory secretions [13, 14], and the importance of prolonged close contact for horizontal transmission is evident in several studies reviewed by Blacklaws *et al.* (2004) [15]. For example, when naïve female goats were kept and milked with SRLV-infected does for 3–5 months in routine dairy conditions, 60% had seroconverted 10 months later [16]. In contrast, five SRLV-free goats pastured with 35–80 SRLV-positive goats had not seroconverted when tested after 7, 9 or 12 months' exposure, although when two were re-tested after 22 months they had seroconverted. Houwers and van der Molen (1987) [17] observed disease progression in a flock of ewes kept under farm conditions for five years, and noted that most seroconversions occurred in summer, implying that transmission had occurred during the winter housing and lambing period. Peterson et al. (2007) [18] kept groups of non-pregnant sheep of two breeds separately at grass for a year, and found that the spread of SRLV from the 9–10 infected ewes to the 19–20 ewes that were initially seronegative in each group was strongly associated with a period of housing that lasted only two weeks in autumn. Several comparisons across Spanish flocks of sheep show a strong association between duration of housing and flock SRLV seroprevalence, and suggest that poor ventilation and flock size are also risk factors [19–22]. The low seroprevalence in extensively-kept sheep flocks that were seldom housed suggested that pasture-based systems of husbandry may offer conditions for the simple and inexpensive control of MV [20]. To test that proposition, we first set out to quantify the rates of SRLV transmission under field and housed conditions, using three longitudinal datasets [17, 18].

There has been almost no quantitative analysis of SRLV epidemiology, nor even estimation of the rate of SRLV transmission under any management regime. A key epidemiological parameter used in the control and eradication of any infectious disease is the basic reproduction number, $R_0$, defined as the expected number of secondary cases which one case would produce in a completely susceptible population. There has been no attempt to estimate its value for SRLV. Cryptic disease syndromes, such as occur from infection by SRLV, pose

particular difficulties for the estimation of transmission rates because the timing and number of infections are obscured by variable delays in seroconversion, unknown delays between infection and infectiousness (latency), and other possible influences such as variation in host susceptibility, environmental factors and co-infections. Quantification of these parameters is a prerequisite for the type of infectious disease modelling that underpins many aspects of modern epidemiological research and the prevention, control and eradication of many human and livestock diseases [23–25]. We applied our parameter estimates to typical commercial flocks to examine under what husbandry conditions virus would spread within a flock and how quickly. Our results provide quantitative evidence on how commercial flocks should be managed in order to control and eradicate MV, and provide new insights into lentivirus biology, transmission and control.

## Methods

### Data sources

Houwers and van der Molen (1987) [17] (hereinafter Houwers) observed a flock of Texel ewes under normal farm conditions for five years from March 1980, starting with 19 uninfected ewes and two infected donors. The flock was kept at grass and allowed free access to shelter in a straw-bedded shed from December to March, where they were offered winter feeding, and where they were kept during nights, and all day during bad weather and throughout lambing in March. Serology was monitored every 5–6 weeks in years 1–3, and less frequently thereafter, by indirect ELISA (100% specificity but unstated sensitivity [26]). The dataset was obtained from [17] and is reproduced in S1 File. Ewes were mated each year by rams from uninfected flocks. MV-infected ewes were kept as long as possible and their ewe lambs preferentially retained as flock replacements. Some other ewes were removed to maintain flock size in the range from 21–37 over the experiment (Fig 1A), yielding records on 45 ewes. Twenty-four of the 43 recipient ewes seroconverted. Four ewes born in 1982 were removed at 8 months, and remained seronegative at 24 months.

In the winter of 1982 the body condition of all ewes deteriorated due to unexpectedly low forage quality. This coincided with a severe infestation of the biting insect *Melophagus ovinus* in early spring and summer. This was associated with a spike in the seroconversion rate in March 1982 with six simultaneous seroconversions (Fig 1B).

The second data source is from an unpublished study by one of the authors [18] who performed two concurrent, 63-week transmission experiments with separate groups of non-pregnant Texel and Blessumer ewes kept at grass for all but 2 weeks, when they were housed. The dataset is reproduced in S2 File. Serology was monitored roughly every two weeks by indirect ELISA (ELITEST-MVV, Hyphen BioMed; specificity 99.3%, sensitivity 99.4%). In the Texel experiment, 10 infected donor ewes were mixed with 20 uninfected recipient ewes. They were housed during weeks 20–21, and 5 recipient ewes subsequently seroconverted over the next 11 weeks (Fig 2A). In the Blessumer experiment, 9 infected donors were mixed with 19 uninfected recipients. They were housed during weeks 18–19; 3 recipient ewes simultaneously seroconverted in the first week of housing and 6 more seroconverted over the next 12 weeks (Fig 2B). At 52 weeks seropositive ewes were no longer tested and were removed from the flock over several weeks for post-mortem examination. Seronegative ewes continued to be tested until the end of the experiment.

### Description of the full mathematical model

We developed a credible model of SRLV transmission and seroconversion of Houwers and Peterson based on infectious disease epidemiology and *a priori* known and unknown aspects

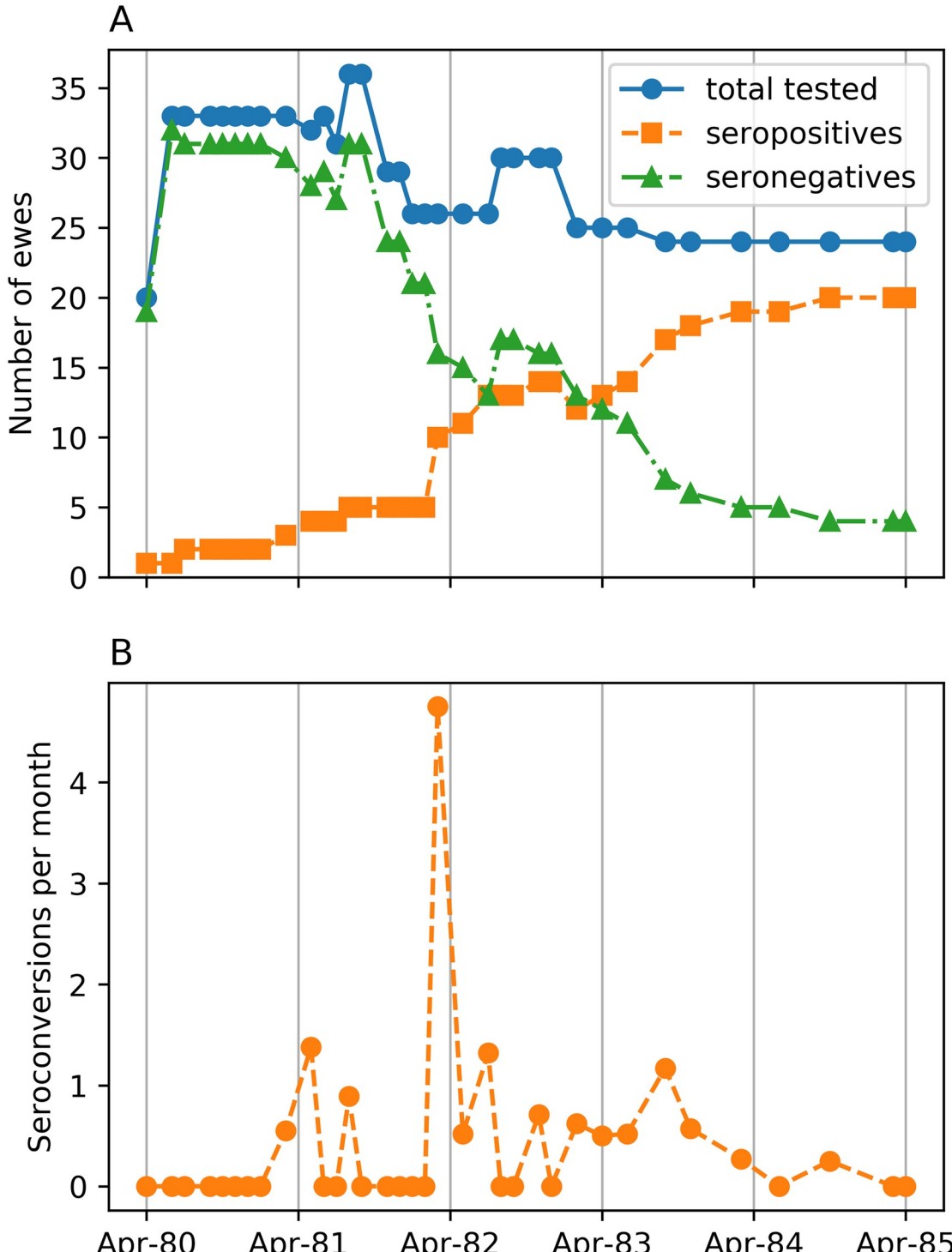

**Fig 1. Houwers' experimental data.** (A) The number of seropositive, seronegative and total ewes tested on each sampling occasion and (B) seroconversions per month of Houwers. (Note: values are not integers because of conversion to a monthly rate).

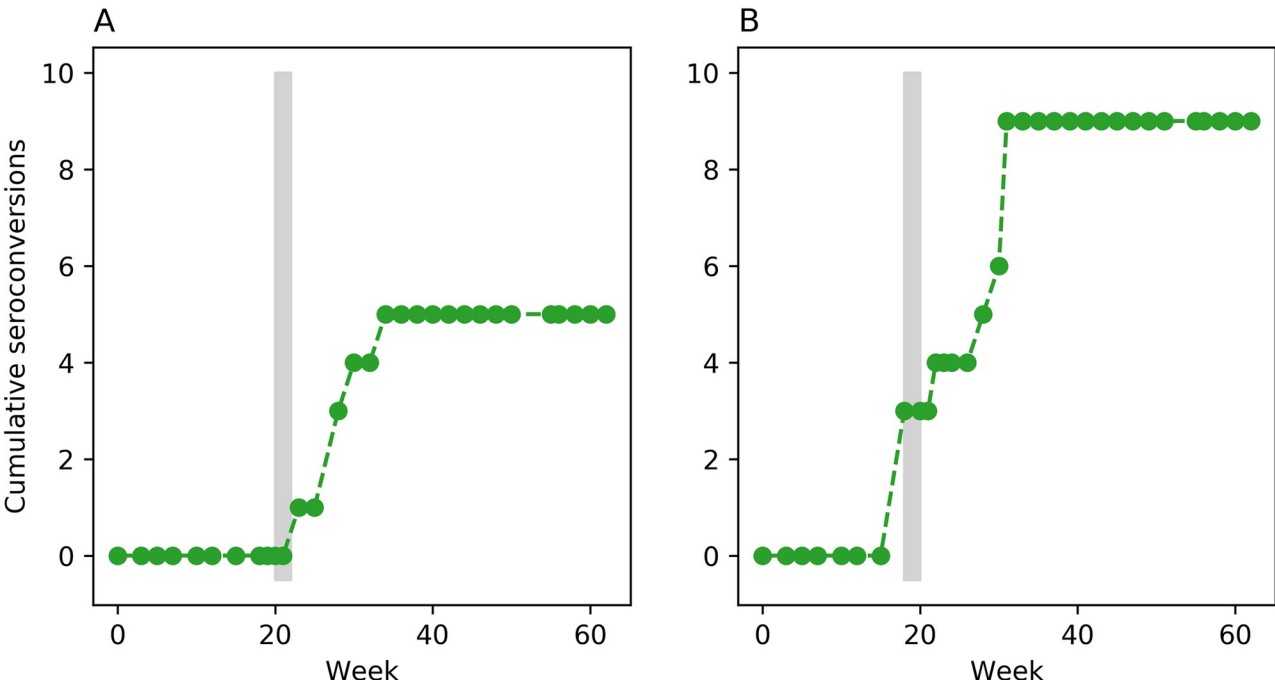

**Fig 2. Peterson's experimental data.** Cumulative number of seroconversions in Peterson's Texel ewes (A) and Blessumer ewes (B). Vertical grey bars show the 2-week housing periods.

of SRLV biology. The parameterisation of this full model is given in Table 1, and its mathematical derivation is detailed in the S3 File. The following assumptions underpin the full model.

- No recipient ewes were infected before the experiment, and nor were the rams.

- Ewes mixed freely and so were exposed to the same infectious environment.

- All lambs were born by the end of March in each year.

- Some ewes may have been resistant to infection so never became infected or seroconverted. The number of resistant ewes is estimated from the data. All other ewes are assumed equally susceptible.

**Table 1. Full model parameters.** Optimal model parameters in bold.

| Parameter | Description | Units |
|---|---|---|
| $\beta_{\text{field}}$ | Field transmission rate* | $(\text{ewe·month})^{-1}$ |
| $\beta_{\text{housed}}$ | **Housed transmission rate**\* | $(\text{ewe·month})^{-1}$ |
| $\beta_{\text{unhealthy}}$ | Housed transmission rate between unhealthy ewes | $(\text{ewe·month})^{-1}$ |
| $\tau$ | **Effective duration of housing** | month |
| $l$ | **Mean latent period** | month |
| $\sigma$ | Standard deviation of latent period | month |
| $\alpha$ | **Shape of distribution of seroconversion period** * | |
| $\mu_{\text{healthy}}$ | **Mean seroconversion period of healthy ewes** * | month |
| $\mu_{\text{unhealthy}}$ | **Mean seroconversion period of unhealthy ewes** | month† |

†One month is assumed to be $^{365}/_{12} = 30.4$ days.
*Included in model of Peterson

- Infectious ewes were equally infectious and infectivity remains constant with duration of infection.

- Transmission of the virus was density-dependent [27], i.e., depends on stocking density of ewes. We assume to two independent, per-capita transmission rates, one when ewes were on pasture and one when they were housed.

- We estimate when ewes were housed. There are no records of duration of housing each month. We must therefore estimate a single, constant rate of housing for all months in which ewes had access to housing. We also estimate the effective number of months ewes had access to housing up to and including March (when we know ewes were housed for lambing). For all other months we assume ewes were kept on pasture.

- Latent period—time from infection to infectiousness—is variable and Gamma distributed. The flexible shape of the Gamma distribution, from exponential to approximately normal, means the form of the distribution can, potentially, be inferred from the data.

- Seroconversion period—time from infection to antibodies detectable by ELISA—is variable and Gamma distributed.

- Mean seroconversion period is independent of force of infection, i.e., independent of dose.

In Houwers, one of the two infected donors introduced into the flock at the start of the experiment showed signs of respiratory distress and its ewe lamb, born that year, seroconverted at 3 months of age. We assume that this donor was infectious. The second donor showed no symptoms. We examine whether this donor was either infectious or latent at the start of the experiment.

To account for the six simultaneous seroconversions in March 1982 when all ewes showed poor body condition the full model contains an additional transmission rate and an additional seroconversion rate. We examine whether none, either or both of these parameters are supported by the data.

In Houwers, twenty-four ewe lambs were retained within the flock. Eighteen of these lambs were born to seronegative dams. One was born to a donor ewe and was vertically infected by its dam. We do not model this infection event, but instead assume this lamb was infected at birth. The other five lambs were born to seroconverted dams and seroconverted at least 14 months after birth. Given this long interval between birth and seroconversion, we assume that these lambs were infected horizontally rather than vertically. Maternal antibodies confer protection to lambs in their first months of life [28]. However, sampling was too sparse to tell if lambs born to seroconverted ewes had maternal antibodies. Allowing for waning, maternally-derived protection in the model resulted in no change in model fit or parameter estimates. For simplicity, we therefore assume that all lambs (except the one born to the donor) were susceptible to infection from birth.

For Houwers, we discretise time into 1-month intervals, so treating per-capita transmission rate, number of infectious ewes, force of infection and seroconversion rate as piecewise constant.

If a susceptible ewe were to be infected at time $T$ and had a seroconversion period of $X$ then it would seroconvert at time $S = T + X$. However, as these times are unknown they must be treated as random variables with associated probability distributions to be inferred from the data. The distribution of infection time, $T$, is determined from the time-varying force of infection which, in turn, is the product of the time-varying per-capita transmission rate and the time-varying expected number of infectious ewes. Transmission rate is specified by four

constant parameters: per-capita transmission rate in the field or when housed, the duration of winter housing and an increased transmission rate in winter 1982 due to poor body condition. The expected number of infectious ewes is determined by the transmission rate and the latent period parameters. The distribution of seroconversion period $X$ is determined by the shape parameter of the Gamma distribution and the mean seroconversion periods of healthy and unhealthy ewes. The parameters of the full model are given in Table 1.

For Peterson, we assume all of the above except that all donor ewes are infectious from the start of the experiments and that housing was just for the designated two weeks. The experiments were too short to estimate latency and resistance, so we assumed that infected recipient ewes did not become infectious and that there were no resistant ewes. The time interval of this model is 1 week.

## Model fitting

The full model determines (i) for each susceptible ewe that seroconverted the probability of it seroconverting when it did and (ii) for each susceptible ewe that never seroconverted the probability of it not seroconverting before it was removed from the flock or the experiment ended. The product of these probabilities gives the likelihood of the data given the model.

Let $\theta$ be the set of model parameters to estimate (Table 1). Let $\mathcal{S}$ be the set of ewe-identification numbers (IDs, S1 and S2 Files) of susceptible ewes that seroconverted and let $\mathcal{N}$ be the set of IDs of susceptible ewes that never seroconverted. Let $p_i(\theta)$ be the probability of seroconverting-ewe $i$ seroconverting between times $s_{-,i}$ and $s_{+,i}$. Let $q_i(\theta)$ be the probability of non-seroconverting ewe $i$ never seroconverting. The likelihood of the data given the model and its parameters $\mathcal{L}(\theta)$ is the product of these probabilities:

$$\mathcal{L}(\theta) = \prod_{i \in \mathcal{S}} p_i(\theta) \prod_{i \in \mathcal{N}} q_i(\theta) \tag{1}$$

The likelihood is multiplied by the prior distributions of the parameters (see S3 File) to obtain the joint posterior distribution of the parameters. We use an adaptive, population based, Markov chain Monte Carlo method with power posteriors to sample the posterior [29–31]. This method is ideal for sampling complex posteriors of high-dimensional, nonlinear, dynamical systems. The Markov chain has a burn-in of $5 \times 10^5$ samples. Inferences are based on $5 \times 10^5$ samples thinned to 5,000 samples to eliminate auto-correlation between samples. Gelman-Rubin statistics [32] and plots of the Markov chains were examined to ensure satisfactory mixing and convergence (see S3 File). The model fitting code is written in C and is available at github.com/nicksavill/maedi-visna-epidemiology.

## Estimating the number of resistant ewes in Houwers

In Houwers, 19 of the 43 recipient ewes never seroconverted. The orange bars in Fig 3 show the distribution of the number of months non-seroconverters were seronegative until they were either removed or the experiment ended. The blue bars show the number of months seroconverters were seronegative before seroconverting. There are four out-lying ewes (IDs 12-0, R86, 9-8 and 23-1) that remained in the flock for longer than 45 months without seroconverting—over 9 months longer than the longest duration a seroconverting ewe took to seroconvert. Post mortem examination of these four ewes did not detect characteristic MV lesions in lungs or udders of the ewes, leading Houwers and van der Molen (1987) to suggest resistance in them. Given the evidence for a genetic basis for resistance [4–6], it is reasonable to try to estimate the number of resistant ewes in Houwers.

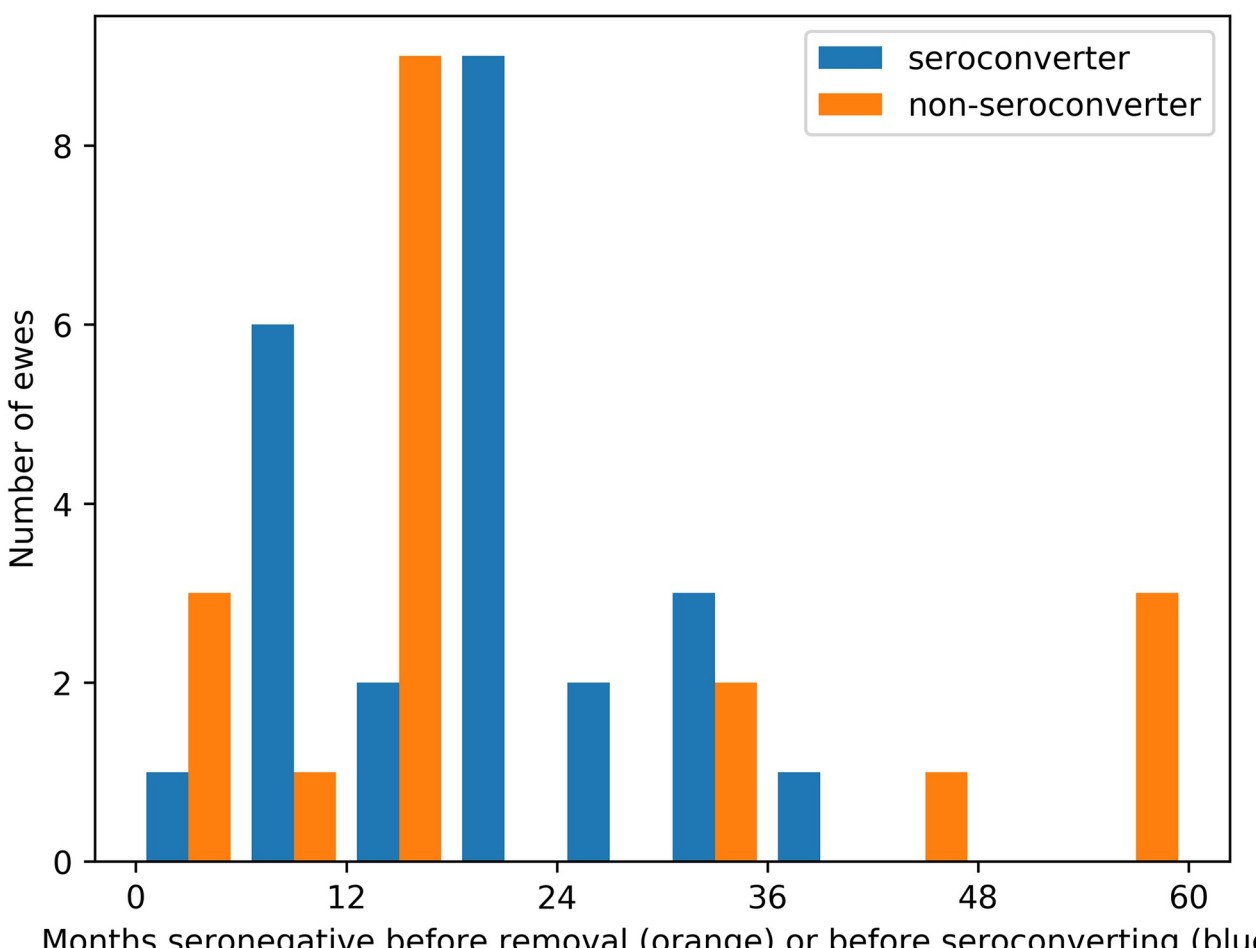

**Fig 3. Seroconverters and non-seroconverters.** Number of months ewes remained seronegative from when they were introduced into the flock until they seroconverted (blue). Orange bars show ewes that did not seroconvert before being removed or the experiment ended.

We cannot use likelihood to determine if non-seroconverters were resistant because the probability of a resistant ewe not seroconverting is 1 by definition. Instead we use the sum of squared residuals (SSR) of the cumulative expected number of seroconversions as a measure of goodness of fit. We test several cases, manually setting which ewes are resistant. For each case we fit the model and calculate SSR. By selecting the case with the lowest SSR as the best fitting model we determine which ewes are likely to be resistant.

### Selection strategy of optimal model and evidence for it

The full model encompasses our *a priori* knowledge and uncertainty of SRLV biology and epidemiology and so may be over-parameterised. A single optimal model that best represents our inferences supported by the data was derived from the full model by excluding different combinations of parameters, refitting these nested models and comparing them using an information-theoretic approach [33]. Akaike's information criterion (AICc) corrected for small sample size [34] is calculated for the full and nested models. AICc rewards goodness of fit (as judged by the likelihood) but penalises additional estimated parameters. The optimal model has the lowest AICc of all models examined. Estimating the number of resistant ewes does not affect the likelihood (see above). In this case the optimal model has the lowest sum of squared

residuals. The optimal model thus provides the reference against which biological hypotheses are tested using model selection criteria [33].

In the Results, we state the optimal model and present and discuss the weight of evidence for each of its components. Relevant parameters of the full model are added or removed from the optimal model and the resulting model refitted to the data. By definition, removal of a parameter from the optimal model results in a worse fit ($\Delta \ln \hat{\mathcal{L}} < 0$). The strength of evidence for the inclusion of the parameter in the optimal model is determined by the formula [35]

$$\text{evidence ratio} = \exp\left(-\frac{1}{2}\Delta \text{AICc}\right) \qquad (2)$$

which quantifies how much less likely a model is relative to the optimal model, and where $\Delta$AICc is the change in AICc relative to the optimal model. Addition of a parameter to the optimal model results in either no better fit or a slightly better fit. In either case, adding the parameter increases AICc relative to the optimal model ($\Delta$AICc $> 0$) and, therefore, indicates that its inclusion is not empirically supported.

## Calculation of $R_0$ and prevalence doubling time

We applied the epidemiological parameters of SRLV to estimate $R_0$ for typical commercial flocks. We use an age-structured model of transmission [36, 37] within a free-mixing group of susceptible ewes of stable size into which a single infected ewe is introduced. The model has 66 age classes each 1 month long to accommodate a latent period of 15 months. We assume that each ewe aged two years and older gives birth to an average of 1.5 lambs, deduct flock losses to natural mortality and culling, and retention of sufficient female lambs to maintain stable group size. We construct a next-generation matrix $A$, whose elements $a_{i,j}$, are the expected number of susceptible ewes of age class $j$ that are infected over the infectious lifespan of a ewe infected at age $i$. Then $R_0$ is the maximum eigenvalue of this matrix numerically calculated using the GNU Scientific library [38].

The prevalence doubling time (for small prevalences) is given by $G\frac{\ln 2}{\ln R_0}$ where $G$ is weighted mean ewe lifespan post infection (see S3 File for details).

## Results

### Evidence for optimal model of Houwers

The optimal model best represents the inference supported by the data [33]. We infer from Houwers (i) resistance in four ewes, (ii) a fivefold increase in seroconversion rate in unhealthy ewes in winter 1982, (iii) density dependent transmission with a negligible field transmission rate and a non-zero housed transmission rate, (iv) a constant, non-zero latent period of roughly 15 months, and (v) a Gamma distributed seroconversion period with a mean of about 8 months (see Figs 4 and 5). The weight of evidence for each of the optimal model's components is given in Table 2.

1. **Resistance in four ewes**. The difference between the sum of squared residuals ($\Delta$SSR) of increasing number of assumed resistant ewes, taken in order of flock residence time without seroconverting, are compared to the optimal model in Table 2 (Models 1–5). The optimal model (with the lowest SSR) suggests that the four non-seroconverting ewes that remained in the flock for over 45 months were resistant (see Fig 3).
   The estimated mean seroconversion period of the optimal model is about 8 months (Table 3). In comparison, the estimated mean seroconversion period if we assume that none of the ewes was resistant (Model 1 in Table 2) is about 25 months. The reason for this

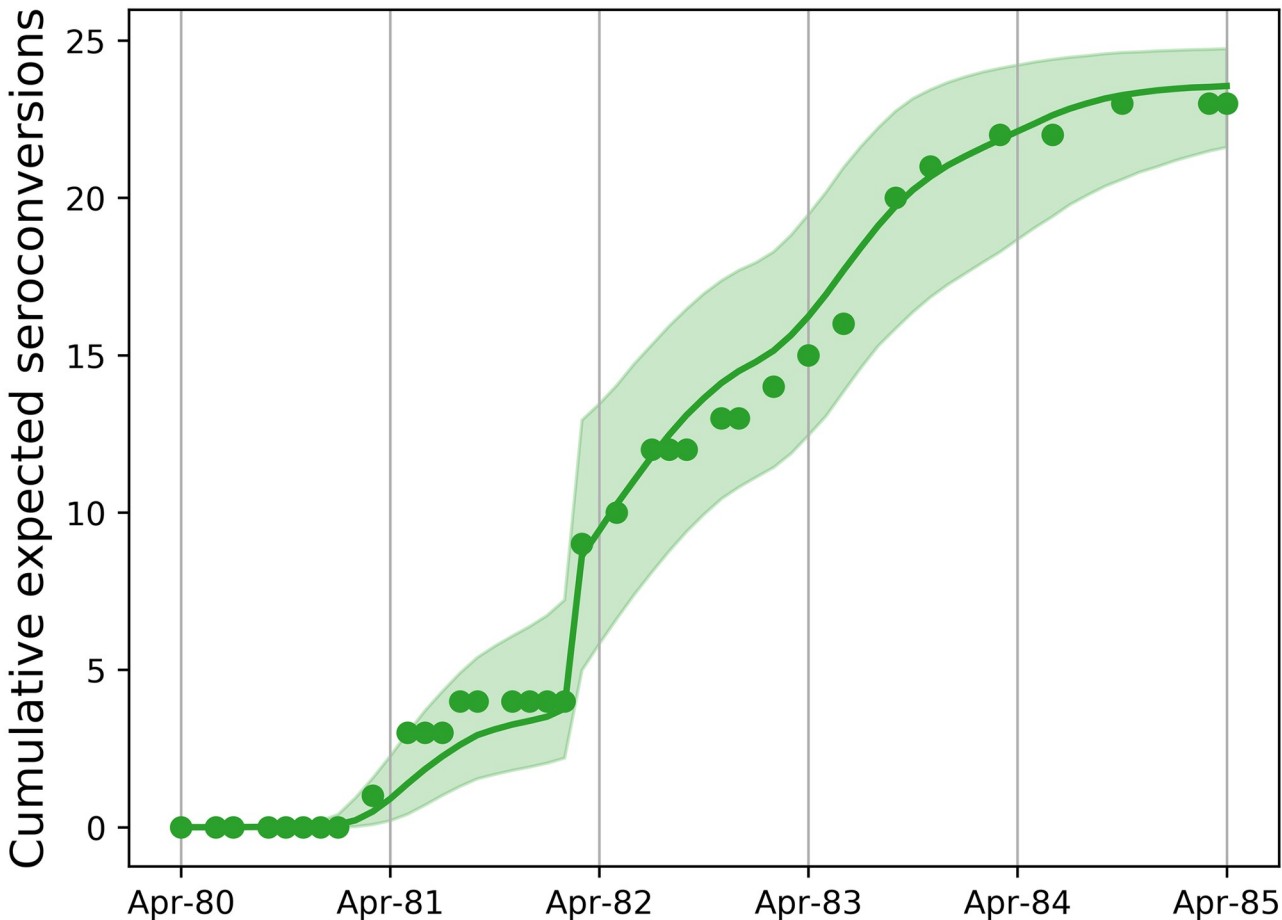

**Fig 4. Model fit to Houwers.** Fit of the optimal model to the cumulative expected number of seroconversions. Circles: data, green line: median model fit: green region: 95% CI.

is that, if the four longest non-seroconverters were susceptible to infection, the probability of them being infected within 4 years can be calculated to be between 0.996 and 1 (95%CI). That is, it is highly probable that, if they were susceptible, they were infected. But to be infected and never seroconvert for over 4 years means the seroconversion period is several years. Such a large mean seroconversion period is not consistent with much of the published evidence, whereas the value of 8 months is. This increases confidence that these four ewes were resistant to infection.

2. **Increased seroconversion rate of unhealthy ewes**. The optimal model suggests that accelerated seroconversion of infected ewes explains the six simultaneous seroconversions in March 1982 when all ewes showed poor body condition. The hypothesis that transmission and seroconversion rates in winter 1982 were no different from other years is rejected (Model 6) having a worse fit than the optimal model ($\Delta \ln \hat{\mathcal{L}} = -6.1$, $\Delta$AICc = 9.5, Table 2) and 114 times less support. The hypothesis that transmission rate was different in winter 1982 compared to other years is rejected (Model 7) having a worse fit than the optimal model ($\Delta \ln \hat{\mathcal{L}} = -6.1$, $\Delta$AICc = 12.3, Table 2) and 480 times less support. Adding an increased transmission rate to the optimal model does not improve fit (Model 8: $\Delta \ln \hat{\mathcal{L}} = 0$) and is therefore not empirically supported.

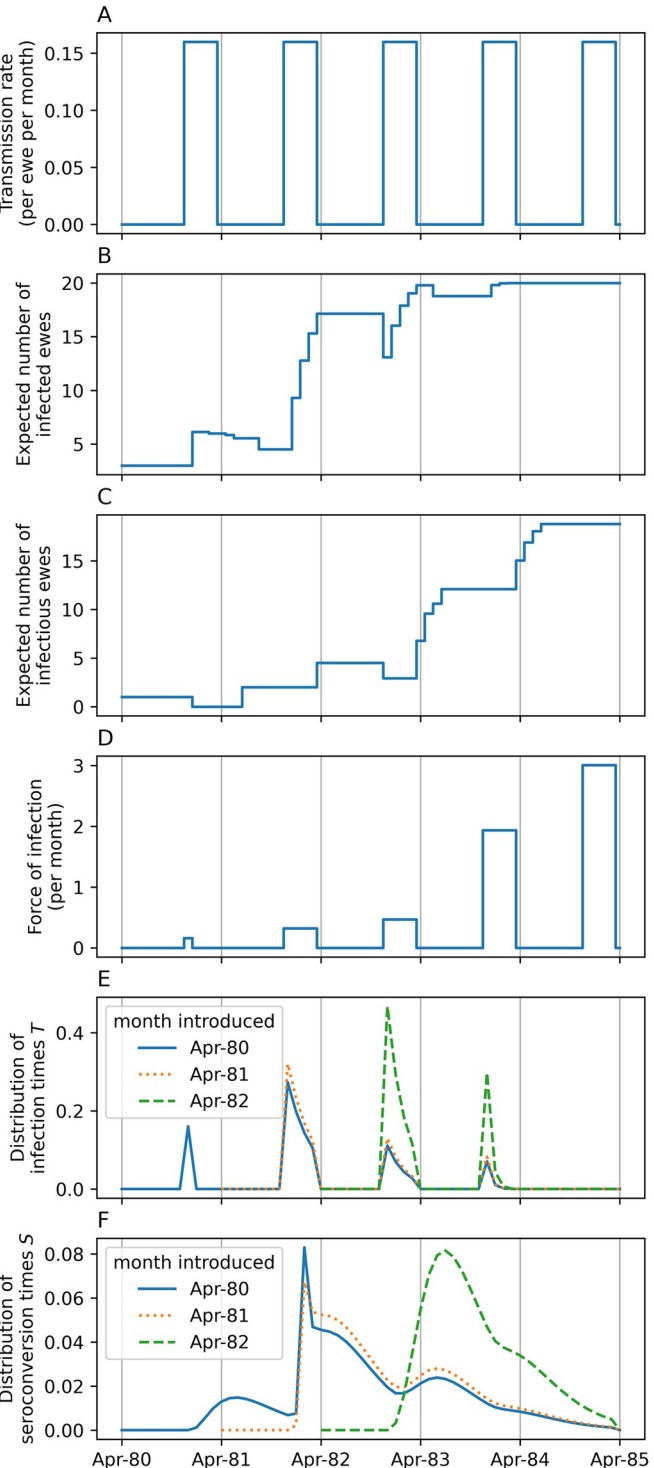

**Fig 5. Optimal model dynamics and predicted distributions.** The dynamics of (A) transmission rate, (B) expected number of infected and (C) infectious ewes for the mean parameter estimates of the optimal model (Table 3, but with $\beta_{\text{field}} = 0$). (D) The force of infection is the product of transmission rate and number of infectious ewes. From this is derived (E) the distributions of infection times $T$, and (F) the distributions of seroconversion times $S$, for ewes introduced into the flock at different times.

**Table 2. Evidence for the components of optimal model of Houwers.** Each model's difference of sum of squared residuals (ΔSSR) or difference of maximum log-likelihood (Δ ln $\hat{\mathcal{L}}$), AICc (ΔAICc) and evidence ratio are calculated with respect to the optimal model.

| Model | | | | Number of parameters | ΔSSR | Δ ln $\hat{\mathcal{L}}$ | ΔAICc | evidence ratio |
|---|---|---|---|---|---|---|---|---|
| **Resistance in four ewes** | | | | | | | | |
| | Minimum number of months a non-seroconverting ewe was resident in flock | Number of resistant ewes | | | | | | |
| 1 | No minimum | 0 | | 6 | 66.5 | | | |
| 2 | 60 | 2 | | 6 | 46.0 | | | |
| 3 | 57 | 3 | | 6 | 40.1 | | | |
| Optimal | 45 | 4 | | 6 | 0 | | | |
| 4 | 32 | 6 | | 6 | 15.4 | | | |
| 5 | 17 | 12 | | 6 | 48.4 | | | |
| **Increased seroconversion rate of unhealthy ewes in winter 1982** | | | | | | | | |
| | | $\beta_{\text{unhealthy}}$ | $\mu_{\text{unhealthy}}$ | | | | | |
| Optimal | Increased seroconversion rate | $= \beta_{\text{housed}}$ | independent | 6 | | | | |
| 6 | No increase in rates | $= \beta_{\text{housed}}$ | $= \mu_{\text{healthy}}$ | 5 | | -6.1 | 9.5 | 1:114 |
| 7 | Increased transmission rate | independent | $= \mu_{\text{healthy}}$ | 6 | | -6.1 | 12.3 | 1:480 |
| 8 | Increased seroconversion and transmission rates | independent | independent | 7 | | 0 | 2.9 | 1:4.2 |
| **Negligible field transmission** | | | | | | | | |
| | | $\beta_{\text{field}}$ | | | | | | |
| Optimal | No field transmission | $= 0$ | | 6 | | | | |
| 9 | Independent field and housed transmission rates | independent | | 7 | | 0 | 2.9 | 1:4.2 |
| 10 | Equal field and housed transmission rates | $= \beta_{\text{housed}}$ | | 6 | | -1.6 | 3.2 | 1:4.8 |
| **Constant latent period** | | | | | | | | |
| | | $l$ | $\sigma$ | | | | | |
| Optimal | Constant latent period | independent | $= 0$ | 6 | | | | |
| 11 | Variable latent period | independent | independent | 7 | | 0 | 2.9 | 1:4.2 |
| 12 | No latent period | $= 0$ | $= 0$ | 5 | | -2.9 | 3.1 | 1:4.8 |
| **Gamma distributed seroconversion period** | | | | | | | | |
| | | $\alpha$ | | | | | | |
| Optimal | Gamma distribution | independent | | 6 | | | | |
| 13 | Exponential distribution | $= 1$ | | 5 | | -1.5 | 0.3 | 1.2 |

The mean seroconversion period of healthy ewes is about 8 months and for ewes with poor body condition is about 1.5 months (Table 3). We can therefore conclude that the six simultaneous seroconversions in infected ewes in winter 1982 were probably due to the ewes' poor body condition causing five times faster seroconversion.

3. **Negligible field transmission**. The optimal model has density-dependent transmission with zero transmission in the field and non-zero transmission in housing. Including a non-zero field transmission rate does not improve model fit and is not empirically supported (Model 9: Δ ln $\hat{\mathcal{L}} = 0$). Equating field and housed transmission rates, so that transmission becomes frequency dependent, gives a slightly worse fit (Model 10: Δ ln $\hat{\mathcal{L}} = -1.6$) resulting in 5 times less support than the optimal model. That support for the optimal model is not higher is due to the effect of the high number of rapid seroconversions of poor condition ewes in winter 1982 masking the signal of housed transmission events in that year. The estimate of field transmission rate, when included in Model 9, is about three orders of magnitude smaller than the housed transmission rate (Table 3). We therefore cannot

**Table 3. Parameter estimates for Houwers and Peterson.**

| Parameter | Mean estimate | 95% CI |
|---|---|---|
| **Houwers** | | |
| $\beta_{\text{field}}$, field transmission rate | $7.5 \times 10^{-4}$ | $2 \times 10^{-5} - 2.3 \times 10^{-3}$ |
| $\beta_{\text{housed}}$, housed transmission rate | 0.17 | $0.04 - 0.39$ |
| $\tau$, effective housing duration | 3.8 | $2 - 4$ |
| $l$, latent period | 15.1 | $2.3 - 21.7$ |
| $\alpha$, seroconversion distribution shape | 3.2 | $0.8 - 8.8$ |
| $\mu_{\text{healthy}}$, healthy seroconversion period | 8.3 | $4.1 - 17.6$ |
| $\mu_{\text{unhealthy}}$, unhealthy seroconversion period | 1.5 | $0.2 - 4.9$ |
| **Peterson, Texels** | | |
| $\beta_{\text{field}}$ | $2.1 \times 10^{-4}$ | $6 \times 10^{-6} - 7.2 \times 10^{-4}$ |
| $\beta_{\text{housed}}$ | 0.09 | $0.03 - 0.22$ |
| $\alpha$ | 1.9 | $0.3 - 6.5$ |
| $\mu$ | 4.6 | $1.1 - 18$ |
| **Peterson, All Blessumers** | | |
| $\beta_{\text{field}}$ | $1.9 \times 10^{-3}$ | $5 \times 10^{-4} - 4.3 \times 10^{-3}$ |
| $\beta_{\text{housed}}$ | 0.11 | $0.04 - 0.21$ |
| $\alpha$ | 6.1 | $0.9 - 18$ |
| $\mu$ | 2.6 | $1.5 - 4.4$ |
| **Peterson, Blessumers without the first three seroconverters** | | |
| $\beta_{\text{field}}$ | $1.1 \times 10^{-4}$ | $4 \times 10^{-6} - 3.3 \times 10^{-4}$ |
| $\beta_{\text{housed}}$ | 0.12 | $0.05 - 0.24$ |
| $\alpha$ | 3.9 | $0.7 - 10.3$ |
| $\mu$ | 3.2 | $1.7 - 9.3$ |

conclude that field transmission rate is actually zero, merely that the model fit is not improved by a non-zero estimate of transmission. This suggests that transmission was negligible when ewes were on pasture, i.e., all transmission events occurred whilst ewes were housed, and therefore at higher density.

4. **Constant latent period**. The optimal model has a constant, non-zero latent period. A variable latent period does not improve model fit (Model 11: $\Delta \ln \hat{\mathcal{L}} = 0$) and is not empirically supported. The marginal posterior distribution of the standard deviation of latent period, $\sigma$, is identical to its prior distribution. We therefore conclude that there is insufficient information in the data to estimate variability in latent period. A non-existent latent period gives a worse fit (Model 12: $\Delta \ln \hat{\mathcal{L}} = -2.9$) resulting in 5 times less support than the optimal model. That support for the optimal model is not higher is due to insufficient information in the data resulting in an imprecise estimate of latent period (2.3–21.7 months, Table 3).

5. **Gamma distribution seroconversion period**. The optimal model has a Gamma distributed seroconversion period. However, the evidence in its favour is not strong (1.2 times more likely) compared to an exponentially distributed period which has one less parameter (Model 13).

One donor was clearly infectious at the start of the experiment because it immediately infected its lamb. The optimal model has the second donor as latent at the start of the experiment. If the second donor is infectious at the start of the experiment the model fit is slightly

worse ($\Delta \ln \hat{\mathcal{L}} = -1.2$) although the evidence is not strong either way with the optimal model is 3.4 times more likely.

## Parameter estimates of Houwers

The mean and 95% CI parameter estimates of Houwers are given in Table 3. The field transmission rate is about three orders of magnitude smaller than the housed transmission rate. Field transmission in Houwers probably had a negligible effect on disease transmission. Although we cannot rule out that its value is zero, we can place an approximate upper limit on its value of 0.002(ewe·month)$^{-1}$ based on the 95% CI.

The effective number of months that ewes had access to housing $\tau$, is estimated to be 3.8 months, suggesting that transmission occurred from early December to March and was not focused primarily during March lambing.

Latent period $l$, is estimated to be just over 1 year. As discussed above, although it is likely that latent period varies between ewes, there is insufficient information in the data to estimate its variability.

Seroconversion period is estimated to be Gamma distributed with shape $\alpha$, of about 3.2. A shape parameter greater than one means that ewes cannot seroconvert immediately after infection.

The estimated seroconversion period of healthy ewes is about 8 months and of ewes with poor body condition it is just 1.5 months. Given the assumption that seroconversion period is Gamma distributed, this means that it may take roughly 1.5 years for 95% of healthy ewes to seroconvert and 2.5 months for 95% of ewes with poor body condition. (These values are found using the Gamma distribution's quantile function at 0.95 with the optimal model's mean parameter estimates).

## Model fits and parameter estimates of Peterson

Only four parameters are fitted to Peterson's data: field and housed transmission rates and shape and mean of the seroconversion period. Recipient ewes that seroconverted are assumed never to become infectious. This is because all seroconversions occurred within 12 weeks of housing which strongly suggests that there were no second-generation transmission events which, in turn, precludes our ability to estimate a latent period. It is also impossible to test if any non-seroconverters were resistant. We therefore assume that all recipient ewes are susceptible.

The data show that seroconversions were almost exclusively associated with the period after housing (Fig 6). The model fit to the cumulative seroconversions in the Texel experiment shows good agreement with the data (Fig 6A). The model fit to the Blessumer experiment is poor (Fig 6B) due to the three ewes that simultaneously seroconverted in the first week of housing. Their infection and seroconversion seem anomalous because no other infections occurred during grazing. We therefore re-estimated model parameters after removing these three outlying ewes from the data. The model fit is substantially improved (Fig 6C). In the following comparison of parameter estimates we use the model without the three anomalous ewes.

As for Houwers, the optimal model of Peterson has zero field transmission rate. A non-zero $\beta_{\text{field}}$ does not change the fit of the model for the Texels (Table 4, Model 1: $\Delta \ln \hat{\mathcal{L}} = 0$) and barely improves it for the Blessumers (Model 1: $\Delta \ln \hat{\mathcal{L}} = 0.6$). The optimal model is overwhelmingly supported over the model of equal field and housed transmission rates (i.e., frequency dependent transmission, Table 4, Model 2: evidence ratios of 1:190 and 1:309),

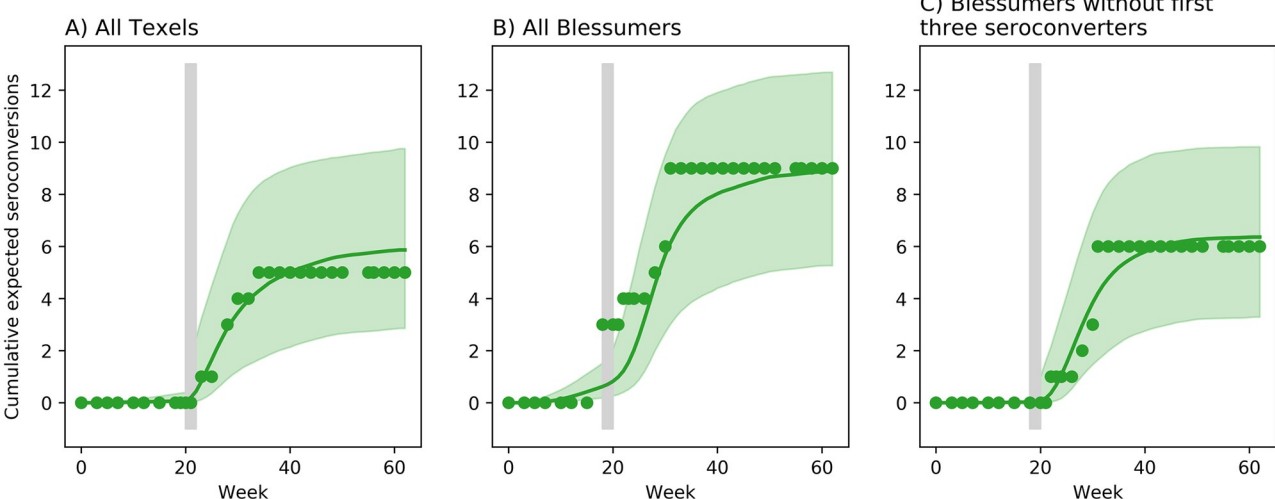

**Fig 6. Optimal model fits to Peterson.** Optimal model fits to cumulative number of seroconversions of Peterson's experiments. (A) Texel ewes, (B) all Blessumer ewes, and (C) Blessumer ewes without the three outlying ewes that seroconverted during housing. Circles: data, green line: median model fit: green region: 95% CI. Vertical grey bars show housing period of two weeks.

supporting our inference from Houwers. The estimated field transmission rates of the full model give upper bounds to this rate. The estimates for $\beta_{\text{housed}}$ are similar for Texels and Blessumers and just slightly lower than for Houwers. Their 95% CIs show extensive overlap. The estimates for $\alpha$ are also similar across the experiments with extensive overlap of the 95% CIs. The estimates for mean seroconversion period $\mu$, in both Texels and Blessumers are about half that in Houwers; about 4 months compared with 8 months. Four months is at the lower end of the 95% CI in Houwers. In the model fit with the three anomalous Blessumer ewes, field transmission rate is biased upwards in order to account for their immediate seroconversion when moving from field to housed conditions (see Discussion).

## Estimates of $R_0$ and prevalence doubling time

Using the parameter estimates we now examine their effect in typical commercial flocks. The very high risk of housing is demonstrated in Fig 7A. Housing groups of 10 susceptible ewes for just 9 days per year is enough for $R_0$ to be above 1, the threshold for persistence of an infection.

**Table 4. Evidence of optimal model of Peterson.** Each model's difference of maximum log-likelihood ($\Delta \ln \hat{\mathcal{L}}$), AICc ($\Delta$AICc) and evidence ratio with respect to the optimal model.

| Model | | | Number of parameters | $\Delta \ln \hat{\mathcal{L}}$ | $\Delta$AICc | evidence ratio |
|---|---|---|---|---|---|---|
| **Negligible field transmission in Texels** | | | | | | |
| | | $\beta_{\text{field}}$ | | | | |
| Optimal | No field transmission | $= 0$ | 3 | | | |
| 1 | Independent field and housed transmission rates | independent | 4 | 0 | 3.2 | 1:4.9 |
| 2 | Equal field and housed transmission rates | $= \beta_{\text{housed}}$ | 3 | -5.2 | 10.5 | 1:190 |
| **Negligible field transmission in Blessumers** | | | | | | |
| | | $\beta_{\text{field}}$ | | | | |
| Optimal | No field transmission | $= 0$ | 3 | | | |
| 1 | Independent field and housed transmission rates | independent | 4 | 0.6 | 2.5 | 1:3.5 |
| 2 | Equal field and housed transmission rates | $= \beta_{\text{housed}}$ | 3 | -5.7 | 11.5 | 1:309 |

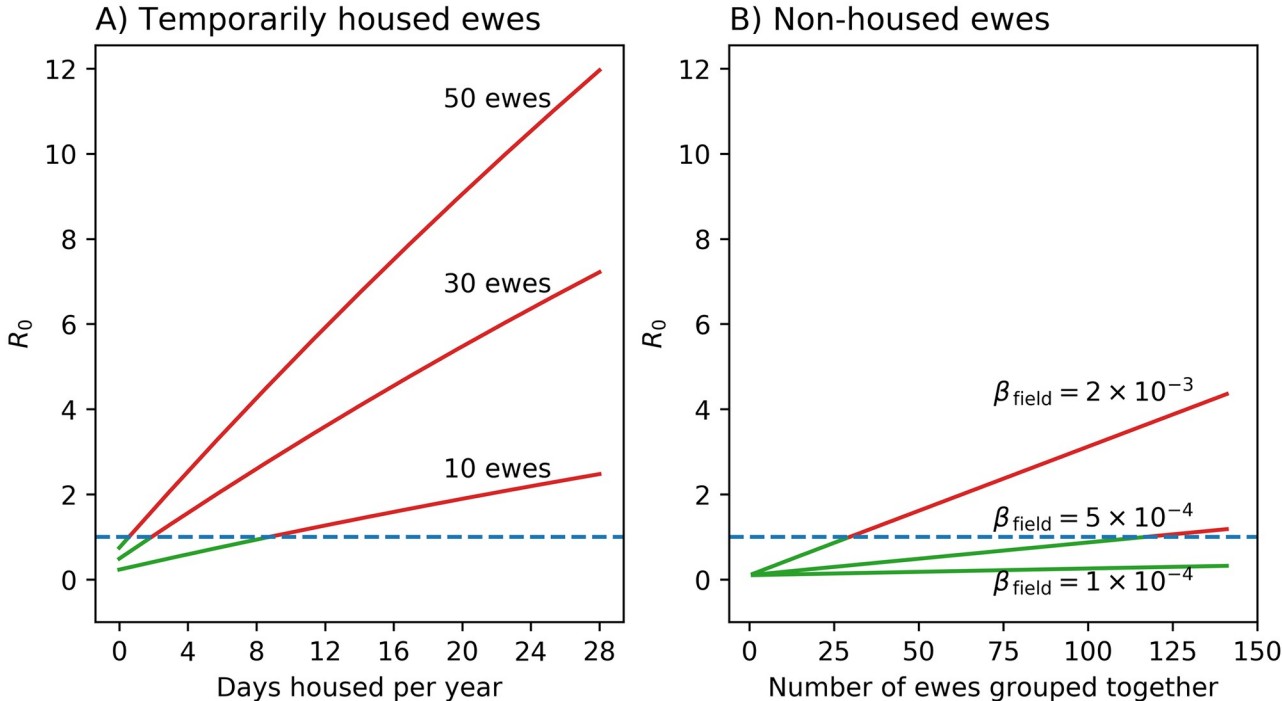

**Fig 7. Factors affecting $R_0$.** (A) Effect of number of days housed per year on $R_0$ for different numbers of susceptible ewes kept together ($\beta_{\text{field}} = 7.5 \times 10^{-4}(\text{ewe·month})^{-1}$, $\beta_{\text{housed}} = 0.17(\text{ewe·month})^{-1}$, $l = 15$ months, maximum age of ewes is 5 years). (B) Effect of field transmission rate on $R_0$ for non-housed groups of susceptible ewes ($l = 15$ months, maximum age is 5 years). All cases have a probability of maternal transmission of 0.1 [14].

For larger group sizes the situation is worse: for example, to prevent transmission in a free-mixing group of 50 susceptible ewes they must be housed for less than 1 day per year. For commercial flocks which house large groups of ewes together for several months over winter and for lambing, $R_0$ is predicted to exceed 10.

In Fig 7B ewes are never housed, and we show the effect of group size and field transmission rate on $R_0$. For a values of $\beta_{\text{field}}$ of $10^{-4}(\text{ewe·month})^{-1}$, close to estimates in Peterson, $R_0$ is well below 1 for typical group sizes. For a value of $5 \times 10^{-4}(\text{ewe·month})^{-1}$, close to the upper 95% CI in Peterson and the mean estimate in Houwers, a group size of less than 100 susceptible ewes is sufficient to keep $R_0$ less than 1. For a value of $2 \times 10^{-3}(\text{ewe·month})^{-1}$, at the upper 95% CI in Houwers, $R_0$ is above 1 for about 25 ewes.

The relationship between $R_0$ and prevalence doubling time (for low prevalences) is shown in Fig 8. For commercial flocks, with predicted $R_0$ greater than 10, doubling time is expected to be about 6 months. As $R_0$ tends to 1 doubling time rapidly heads into the decades making the possibility of outbreaks negligible and the chance of infected ewes being removed by demographic stochasticity high.

## Discussion

Quantitative analysis of infectious disease control measures depends critically on accurate estimates of epidemiological parameters. Yet the biological processes resulting in rates of disease transmission, seroconversion delays and latent periods are inherently stochastic, and may vary with host state and environmental conditions. We combined comprehensive, longitudinal serological data with a dynamical mathematical model to estimate key epidemiological parameters of SRLV transmission under typical farm conditions and derive, for the first time,

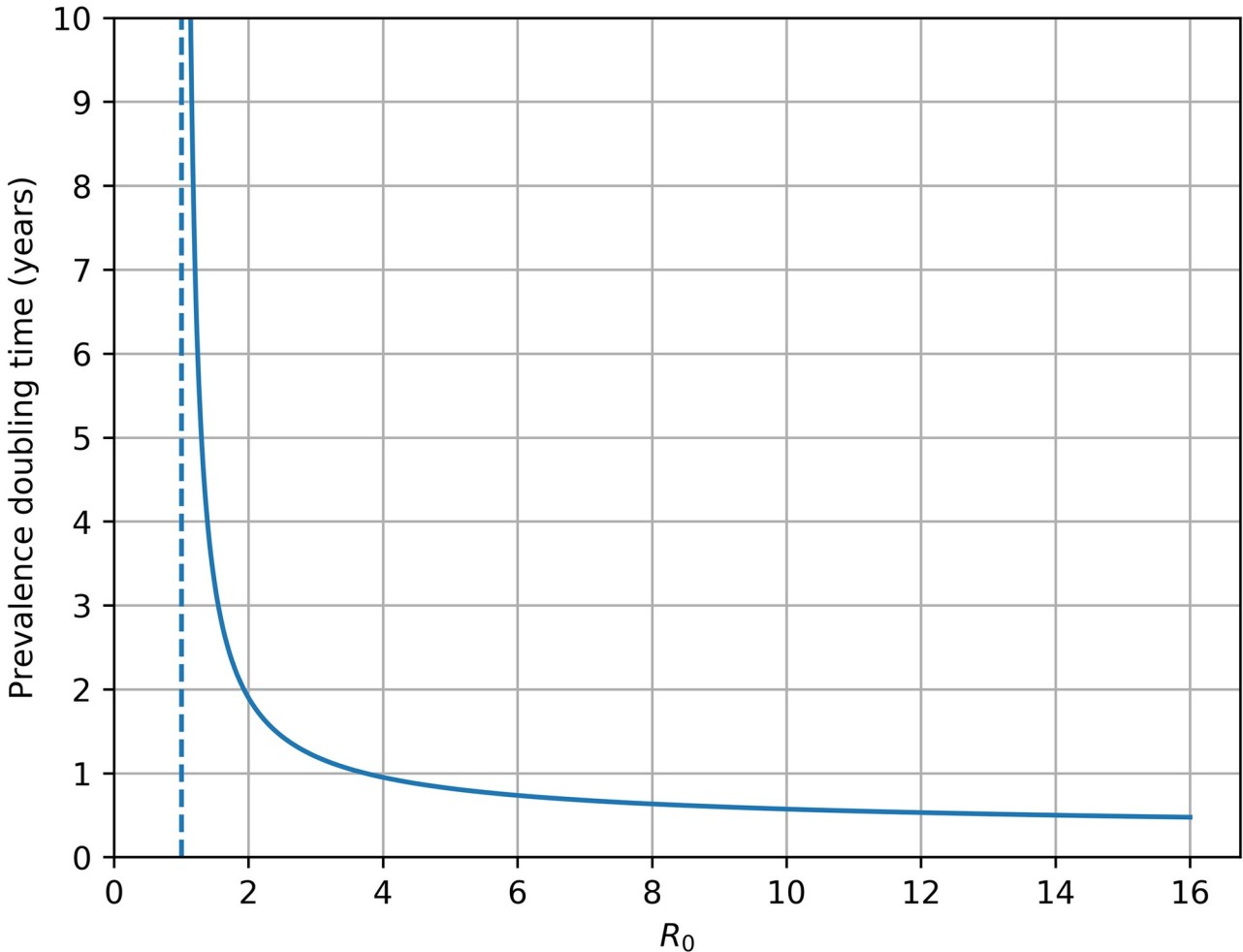

**Fig 8. Prevalence doubling time with $R_0$ at low prevalences.**

predictions of $R_0$ under various durations of housing. The resulting characterisation of the epidemiological process in MV reveals strong density-dependence in transmission rate, seroconversion periods that were markedly shorter in ewes with poor body condition and health, a latent period almost twice as long as seroconversion period and variation in host susceptibility. The finding that SRLV transmission is negligible in pastured animals supports the suggestion of Leginagoikoa *et al.* (2006) [20] that the disease could be controlled simply by avoiding the use of housing for anything but short periods.

## Latencies

The delay between infection and seroconversion and its variability are well known (e.g., [9]) but have not been accurately quantified because of the difficulty of knowing when natural infections occur. Laboratory infection by efficient routes produces PCR-detectable virus typically within 1–2 months, and seroconversion follows roughly 1–4 months later [39]. Less efficient routes and lower doses of virus take longer to present detectable virus and cause longer seroconversion delays [40]. In one of the few quantifications of seroconversion delay, the serostatuses of 20 dairy goats, naturally infected and PCR-positive for SRLV, were monitored [9]. Over the following eight months only 10 had seroconverted with a mean delay of 5.1 months

and coefficient of variation of 34%. Our results are consistent with this in normally-healthy ewes, with mean seroconversion delays of 3 and 5 months (Peterson) and 8 months (Houwers) (Table 3). Thus the sequence of events in natural infections appears to be more prolonged than in experimental infections, implying that a feature of the natural infection process is that it is dose-limited.

We found that the mean seroconversion delay was markedly reduced in ewes with poor body condition (1.5 vs. 8.3 months, Table 3), due to poor nutrition and a parasitic infestation. Poor body condition would presumably have reduced the resources available to host immune defences against the virus, allowing it to replicate more rapidly. The widespread puncturing of the skin by parasites would have activated an inflammatory immune response, including circulating cytokines, that could also have augmented the immune reaction to virus, speeding seroconversion. Clinical observation suggests that poor husbandry conditions and the presence of secondary infections are conducive to more rapid spread of MV [41].

It might be expected that sheep become infectious soon after infection given the early viraemic phase 2 to 8 weeks post infection [42] and the presence of PCR-detectable virus before seroconversion (e.g., [39]). We infer from our analysis of Houwers that a latent period exists and is about 15 months, or roughly twice as long as the time from infection to seroconversion (Table 3). The 95% CI of latent period is broad (2.3–21.7 months), suggesting that there is a period of at least a few months following infection, and most likely about a year, during which sheep are not infectious. This novel result warrants further research to improve the precision of its estimate. Delayed SRLV transmission can be inferred from the positive cell-mediated immune response or the presence of virus in previously naïve ewes being found only after 20 months of close proximity with 13 experimentally-infected ewes [43]. Virus was isolated from one of the lambs born to the infected ewes and kept with them about 14 months after experimental infection of the ewes, but not from the other four lambs kept until 9-11 months of age. The existence of a latent period could also be construed from the observation by Torsteinsdóttir *et al.* (2003) [44] that one sheep infected experimentally, and that seroconverted after seven weeks, did not appear to transmit SRLV to four naïve sheep during the six months they were penned together (as judged by a lack of detectable circulating virus or specific antibodies). One possible explanation, following the work of Arnarson *et al.* (2017) [8], is that the virus that is transmitted has to evade the immune response in the host animal by mutation to allow it then to go on to infect other animals, and the six-month period was insufficient for development of antigenic variants required for transmission (Torsteinsdóttir pers. comm.). Furthermore, SRLV must be present in a transmissible form, i.e., in alveolar fluid, and in sufficient quantity, and both of these are contingent on well-developed clinical stages of the disease, which necessarily entails a delay since the time of infection.

### Transmission and housing

A striking finding is that MV transmission was negligible during the grazing period, implying that all infections occurred within periods of housing. Our inferences from Houwers are corroborated and refined by Peterson's data. In the latter, housing of non-pregnant ewes for only two weeks resulted in multiple infections. This shows, incidentally, that pregnancy *per se* is not implicated in transmission risk. The only anomalous result is that three Blessumer ewes seroconverted in the first week of housing, and so must have been infected before being housed. Assuming they were not infected before the experiment, they could have been infected whilst grazing, but that is unlikely given the very clear absence of infections during grazing elsewhere in the Peterson experiments (Fig 6). The gathering of the ewes for blood-sampling every two weeks, during which they were held together in pens for 1–2 hours, clearly presented a small

increased risk of transmission and, together with the breed's reputation for increased suscepti-bility to SRLV, may explain the early seroconversion of some Blessumers. The effect of these ewes on the analysis was to inflate the estimated field transmission rate 10-fold. Without these ewes, all of the Blessumer ewe parameter estimates are consistent with the Texel ewes of Houwers and Peterson (Table 3).

We estimated the transmission rate under grazing to be about $10^{-3}$ to $10^{-4}$(ewe·month)$^{-1}$, compared with about $10^{-1}$(ewe·month)$^{-1}$ during housing. The difference of two to three orders of magnitude corresponds roughly to the difference in stocking densities. The allow-ance of shed space in the Peterson experiment was about 3 m$^2$/ewe, and field stocking rate was about 1,000 m$^2$/ewe. Previous studies have shown an association between seroprevalence and duration of housing across a range of breeds, intensities and purpose of husbandry (meat or milk). For example, Manchega-cross sheep kept almost permanently at pasture had low mean seroprevalence (c. 5%) compared with 25% in semi-intensive Latxa dairy flocks housed for between 2 and 8 months per year and 77% in intensive Assaf dairy flocks housed almost con-tinuously [20]. It could not be shown, however, that duration of housing explained any of the variation within these breeds and production systems.

The high transmission rate during housing emphasises the importance of close proximity: sheep kept at high density are more likely to transmit SRLV. Yet infection risk across even quite small distances is apparently low, with separation distances of 2 m between penned adult goats being sufficient to prevent infection with SRLV [16]. Alveolar fluid of SRLV-infected sheep contains both cell-associated and cell-free virus, and virus can be detected in the envi-ronment of infected sheep penned together and in the air they exhale [45]. Lung fluid is, most obviously, a source of infection when sheep cough. The trachea is a site of virus entry to sheep but is a less efficient one than the lower lung, which has an abundance of potential target cells for SRLV, including alveolar macrophages that clear the lung of inhaled particles [46]. Com-pared with the intratracheal route, higher doses of virus are required for infection to occur via the conjunctival space [40], and intranasal inoculation is considered to be inefficient [44]. It has been reported that it was rare for virus to be detectable in nasal or throat swabs [47]. Thus the major site of SRLV entry during respiratory transmission is the lower respiratory tract and not the nasal cavity or nasopharynx, and it appears that sheep are unlikely to contract SRLV by purely nose-to-nose contact. Twenty-two lambs allowed nose-to-nose contact for nearly 2.5 years with their SRLV-positive dams through a fence but not sharing a drinker showed no infection until 6 years of age, when one maternally-derived infection was detected [14, 48]. There is rather limited information on precisely what aspect of housing—such as the proximity of sheep, the shed's ventilation characteristics, the sharing of drinkers and feeding areas—are most associated with SRLV transmission. Further research could indicate whether re-design-ing sheep housing would have merit in controlling transmission.

Close proximity in housed sheep does not always result in the spread of SRLV, as shown by two experiments on Spanish dairy sheep [49, 50]. Lambs were reared on ovine or bovine colos-trum (O, B). Most were born to seropositive dams, and 6% were LTR-positive at birth. The groups were penned separately until weaning at 5–6 weeks, and then all lambs were regrouped in the two pens according to weight, and this was repeated twice subsequently. The O lambs, which showed evidence of higher SRLV infection (15–55% LTR-positive from 30–300 days), thus posed a horizontal infection risk to the B lambs. Yet, by 300 days the low numbers of B lambs positive for LTR (0–2%) and by ELISA (2.5–10%) suggests that limited, if any, horizon-tal transmission occurred. In a subsequent experiment on these lambs, seroprevalence in the ewe lambs joining the adult dairy flock and housed for 6 months a year rose over the following two years from 15% to 57% (cf. the adult flock's 42–66% seroprevalence over this time) [51]. Yet, comparable male lambs kept together in a separate shed showed no such increase. No

transmission of the SRLV infection between males was detected over the two years. These experiments suggest that transmission between lambs is limited, perhaps because of latency, and show that prolonged close contact need not result in SRLV transmission, at least amongst castrate males. The females were in close contact with adult females with, presumably, well-established SRLV infection and possibly other infections, and all were subject to the metabolic stress of pregnancy, parturition and lactation as well as repeated handling for milking. Coping with these could well have reduced the resources available to the immune control of viral replication.

## Horizontal and vertical transmission

Horizontal transmission must be put in the context of vertical, maternal, transmission in utero, in colostrum or milk, or by the respiratory route directly from dam to offspring. Infected epithelial cells in the mammary gland are a potential source of infection in colostrum [1], and artificial rearing is a standard procedure for establishing uninfected flocks from valuable parent stock [52]. However, more recent work shows that postpartum maternal transmission is not efficient under normal circumstances. Only about 16% of lambs separated and reared away from their SRLV-seropositive dams after 24 hours of natural suckling had seroconverted by 300 days [49]. SRLV provirus and maternal antibodies were efficiently transmitted to newborn lambs fed on colostrum from SRLV-infected ewes, yet none went on to develop persistent infection and seroconvert, possibly because of the co-occurrence of virus and maternal antibodies in colostrum [48]. Of the 22 lambs studied, none were serologically- or provirus-positive over the subsequent 5.5 years, but a follow-up study reported that one of the lambs had seroconverted by 6 years and proviral sequence analysis showed that the infection was from maternal transmission but by an unknown route [14]. The same study investigated true maternal transmission in a further 40 SRLV-infected dam-daughter pairs and it was shown to account for only 10–14% of transmission. The remaining 86–90% of infections therefore resulted from horizontal transmission, almost certainly by the respiratory route.

## $R_0$

$R_0$, the basic reproduction number, is the expected number of secondary cases which one case would produce in a completely susceptible population. An $R_0$ greater than 1 results in infection spreading. Its value is determined by multiple factors, including transmission rate, housing duration, the number of ewes in a group, the latent period and the maximum age of ewes. Housing has by far the greatest impact on $R_0$ due to the large transmission rate when ewes are housed. We show that, even in small group sizes, $R_0$ is greater than 1 when animals are housed for just over one week per year. Therefore outbreaks are almost inevitable in most commercial flocks if infection is present. The speed with which an outbreak proceeds is given by the prevalence doubling time. In housed flocks, which have large $R_0$, the doubling time is roughly 6 months.

Transmission rate on pasture is estimated to be about three orders of magnitude smaller than the housed rate, and we found no evidence that transmission occurred on pasture. Therefore, by not housing ewes, $R_0$ is likely to be below 1 for group sizes of at least up to 100 ewes. Note that, as group size increases, our assumption of well-mixing breaks down, especially over short times—a ewe is unlikely to come into contact with all other ewes in a large group over a few hours or even days. The effect of this is to cause contact rate and, therefore, transmission rate and $R_0$ to plateau at large group sizes held in close proximity for short periods. Flocks with a high prevalence of SRLV that are switched from winter-housing to outdoor lambing, and for

which $R_0$ should be less than 2, even with normal gathering for shearing and other procedures, should see prevalence decline rapidly over the following years of normal flock turnover.

Although we have included maternal transmission in our estimates of $R_0$, its effect is negligible. Only one replacement ewe lamb is needed over the lifetime of the average ewe to maintain flock size, much as one infection is required over an infected animal's lifetime for a disease to persist. If only one of infected ewes' lambs are maternally-infected [14], then the contribution to $R_0$ by this route can only be 0.1, which is a trivial contribution to disease persistence (see [19]).

## Conclusion

MV is a debilitating disease that is costly and reduces animal welfare. Mathematical modelling has allowed us to synthesise hypotheses about MV epidemiology and existing datasets in an established theoretical framework. We have quantified essential epidemiological parameters and applied them to the prediction of management outcomes, and will now be able to examine cost-effective management strategies and implications for testing and accreditation procedures in future work. Elimination and accreditation will always have a part to play in avoiding spread between flocks. But management techniques which can inexpensively minimise disease spread and reduce high prevalence in infected flocks could have an important part to play in commercial flocks. Our research suggests that extensive all-grass systems, which are increasingly being adopted in the UK for economic reasons, could avoid MV being a problem.

The results also add precision to our understanding of the disease process: delayed seroconversion and the period of epidemiological latency are identified as key time delays in the disease process in MV/OPP. Time delays in seroconversion merely make the disease more difficult to detect and control, but the significance of a time delay in infectiousness is that it acts to slow the spread of virus. The gradual build-up of inflammatory lesions, especially in the lungs, eventually triggers seroconversion. Before the animal becomes infectious, there follows a further time delay during which, we infer, advancement of disease presents increasing quantities of infective alveolar macrophages and free virus which are eventually sufficient for transmission. This implies increasing infectiousness as the disease progresses, for which there is tentative evidence [28, 53], but insufficient information in the present data to model.

## Supporting information

**S1 File. Houwers' serological data.** Excel spreadsheet of serological data from [17].
(XLSX)

**S2 File. Peterson's serological data.** Excel spreadsheet of serological data from [18].
(XLSX)

**S3 File. Additional details of methods.** Derivation of the mathematical model, selection of prior distributions, derivation of $R_0$ and prevalence doubling time and assessment of Markov chains of the optimal model.
(PDF)

## Acknowledgments

We thank Katie Atkins for insightful discussions and comments on the paper. We thank Sigurbjörg Torsteinsdóttir for helpful comments on her MVV research. We thank Roland Kao, Andrew Leigh-Brown and John Vipond for comments on the paper. Finally, we thank Dirk Houwers for sharing his experience and knowledge of his experiment with us.

## Author Contributions

**Conceptualization:** Andrew W. Illius, Nicholas J. Savill.

**Data curation:** Karianne Lievaart-Peterson.

**Formal analysis:** Andrew W. Illius, Nicholas J. Savill.

**Investigation:** Andrew W. Illius, Nicholas J. Savill.

**Methodology:** Andrew W. Illius, Nicholas J. Savill.

**Software:** Nicholas J. Savill.

**Validation:** Andrew W. Illius, Karianne Lievaart-Peterson, Tom N. McNeilly, Nicholas J. Savill.

**Visualization:** Andrew W. Illius, Nicholas J. Savill.

**Writing – original draft:** Andrew W. Illius, Nicholas J. Savill.

**Writing – review & editing:** Andrew W. Illius, Karianne Lievaart-Peterson, Tom N. McNeilly, Nicholas J. Savill.

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
