## [Decision Letter · Decision Letter 0]

28 Feb 2020

PONE-D-19-32266

Epidemiology and control of maedi-visna virus: curing the flock

PLOS ONE

Dear Dr. Savill,

Thank you for submitting your manuscript to PLOS ONE. After careful consideration, we feel that it has merit but does not fully meet PLOS ONE’s publication criteria as it currently stands. Therefore, we invite you to submit a revised version of the manuscript that addresses the points raised during the review process.

In addition to the reviewer comments below, please clarify the following points:

Please clarify the ownership of the original Peterson dataset.

Please clarify the ownership of the Houwer dataset.

Please confirm whether the Peterson and Houwer datasets, and any other datasets used in this study, are publicly available in your paper and its Supporting Information files, as per the PLOS ONE data availability policy (https://journals.plos.org/plosone/s/data-availability#loc-introduction). If they are not, please clarify how other researchers may access these datasets, and whether they are publicly available. Please also clarify in your Methods section how the authors obtained these datasets. Can you please also clarify if Dr. Houwer had any contributions to this manuscript?

Please clarify if the manuscript entitled "Quantification of the horizontal transmission of maedi visna virus between sheep in two different breeds" (Peterson K., Stegeman J.A., Colenbrander B., Houwers D.J), reference 12, utilizes the same underlying dataset and if the papers are related submissions. Please note that PLOS ONE requires authors to declare any related manuscripts under consideration at other journals in the cover letter of their submission.

We would appreciate receiving your revised manuscript by Apr 13 2020 11:59PM. To enhance the reproducibility of your results, we recommend that if applicable you deposit your laboratory protocols in protocols.io, where a protocol can be assigned its own identifier (DOI) such that it can be cited independently in the future. For instructions see: http://journals.plos.org/plosone/s/submission-guidelines#loc-laboratory-protocols

We look forward to receiving your revised manuscript.

Kind regards,

Hanna Landenmark, Associate Editor, PLOS ONE

on behalf of

Lucy C. Okell

Academic Editor

PLOS ONE

Journal Requirements:

Reviewers' comments:

Reviewer's Responses to Questions

**Comments to the Author**

1. Is the manuscript technically sound, and do the data support the conclusions?

Reviewer #1: Yes

Reviewer #2: Partly

2. Has the statistical analysis been performed appropriately and rigorously? 

Reviewer #1: Yes

Reviewer #2: I Don't Know

3. Have the authors made all data underlying the findings in their manuscript fully available?

Reviewer #1: Yes

Reviewer #2: Yes

4. Is the manuscript presented in an intelligible fashion and written in standard English?

Reviewer #1: Yes

Reviewer #2: Yes

5. Review Comments to the Author

Reviewer #1: I found this to be an interesting and informative manuscript. However, I do have some concerns about the clarity of the model comparison done that I think should be addressed prior to publication. In addition, there are a number of typos throughout that could have been avoided by the use of a simple spell-check.

By my count, the authors propose 720 potential models that reflect all possible combinations of 5 values of the number of resistant ewes, 2 levels of donor infectiousness, 4 effects of the winter of 1982, 3 levels of latency, 3 forms of transmission, and 2 shapes of the distribution of seroconversion periods (from Table 2). However, they only present results for 20 models. This implies that the authors had some a priori method of selecting from all possible model combinations. This should be described in the text. For example, did the authors select from all parameters using AIC to arrive at a model structure before selecting the number of resistant ewes? If so, what was the number of resistant ewes used to arrive at that initial model? If, as is possible, the values presented in table 2 are marginal AICs over all possible alternative models combining other parameters, then this needs to be described.

L184-5 Authors state that convergence and mixing were assessed. These measures should be presented in the supplement so that readers can assess these on their own.

L188 The authors refer to “hypothesis testing” though really what is being done here is model selection. There are no formal null hypotheses or distributions of test statistics employed. This is a fairly trivial semantic point, but one that I would recommend should be changed. Throughout, the authors are simply choosing the best-fit model according to an information criterion and language used should reflect that.

L229 The authors should state what the preferred model is at this point. Simply stating the AIC makes the remaining comparisons confusing. Per the comment above about the full set of possible models, it is difficult to assess whether differences in individual parameters are the cause of better fit in the model comparisons below or if the bulk of the improved fit is driven by, for example, the change in the number of resistant ewes in the best fit model.

L324-327 This approach to removing 3 ewes to improve fit seems to come out of the blue and appears ad hoc. The authors should provide more justification for this choice. This seems analogous to the treatment of the resistant ewes for which the authors took a more formal approach. Is there not a parallel approach that could be applied here to address the influence of these 3 ewes? (Of course likelihoods are not comparable if you remove these 3 ewes, so one would need some change in parameterization to account for why these ewes are “ignored”).

L509 R0 should be defined earlier in the manuscript, when it is introduced.

L547 “avoid” seems a strong statement here. I would soften this language. Perhaps this research suggests that MV may be a lesser problem, but I don’t think this goes far enough to justify that the problem would be “avoided”.

Table 3 — I would recommend including an comparison of the prior and posterior distributions of the parameters in the best fit model so that readers can assess the information on each of these parameters. It isn’t clear how many of these have moves substantially away from the prior.

Reviewer #2: Interesting piece of work modeling intra flock transmission of SRLV in sheep.

The viruses studied should be referred to as small ruminant lentiviruses and not MVV nor OPPV; it has been more than 20 years that those SRLV were demonstrated as heterogeneous and common to both sheep and goats (ann virol 1997 142: 1125 should be quoted); the interspecies transmission as quoted in ref 1 is misleading.

There is a major discrepancy between the potent mathematical tool and the small size of the 3 experiments of 20 -30 animals each.

The economic consequences of SRL infection quoted as a motto with references to reviews should be backed by research papers that are not existing to my knowledge.

The horizontal transmission has been suggested as early as beginning of this century (see Vet Res 2004 35: 257; paper that review correctly SRLV infection.

There is a strong issue regarding the property of the original datas also the so called Peterson original set of data seems accessible this is not the case of the so called houwers dataset. The origin, accessibility and ownership of the original experimental datas should be stated. Reading through dr Peterson thesis it seems she intended to publish a similar analysis to the present manuscript based on what is her own results . What is the status of this manuscript (Peterson K., Stegeman J.A., Colenbrander B., Houwers D.J. Quantification of the horizontal transmission of maedi visna virus between sheep in two different breeds. Submitted for publication) see ref 12 p127 of the thesis. Furthermore since all 3 experiments are from dr Houwers team how come he is not a coauthor. The reviewer has major concern regarding appropriation of datas and ideas.

The conclusions are not backed by the observations and the analysis, the experimental conditions of the 3 flocks were outside and inside.

The concept of non infected animals is naive when seronegative animals can be µPCR positive in the blood or in the semen (see Therionelogy 2008 69: 433. Thus the first assumption (namely that the rams were not infected) is not supported.

6. PLOS authors have the option to publish the peer review history of their article (what does this mean?). If published, this will include your full peer review and any attached files.

Reviewer #1: No

Reviewer #2: No

---

## [Author Response · Author response to Decision Letter 0]

17 Mar 2020

Specific editor comments have been addressed in our cover letter

---

## [Decision Letter · Decision Letter 1]

10 Jul 2020

PONE-D-19-32266R1

Epidemiology and control of maedi-visna virus: curing the flock

PLOS ONE

Dear Dr. Savill,

Thank you for submitting your manuscript to PLOS ONE. After careful consideration, we feel that it has merit but does not fully meet PLOS ONE’s publication criteria as it currently stands. Therefore, we invite you to submit a revised version of the manuscript that addresses the points raised during the review process.

We look forward to receiving your revised manuscript.

Kind regards,

Stephen Raverty

Academic Editor

PLOS ONE

Additional Editor Comments (if provided):

Thank you for the revised version of the manuscript. The text is considerably improved. However, a third reviewer has provided a comprehensive assessment of the most current version of the paper and these comments should be addressed prior to publication.

Reviewers' comments:

Reviewer's Responses to Questions

**Comments to the Author**

1. If the authors have adequately addressed your comments raised in a previous round of review and you feel that this manuscript is now acceptable for publication, you may indicate that here to bypass the “Comments to the Author” section, enter your conflict of interest statement in the “Confidential to Editor” section, and submit your "Accept" recommendation.

Reviewer #2: All comments have been addressed

Reviewer #3: (No Response)

2. Is the manuscript technically sound, and do the data support the conclusions?

Reviewer #2: Partly

Reviewer #3: Yes

3. Has the statistical analysis been performed appropriately and rigorously? 

Reviewer #2: Yes

Reviewer #3: Yes

4. Have the authors made all data underlying the findings in their manuscript fully available?

Reviewer #2: Yes

Reviewer #3: Yes

5. Is the manuscript presented in an intelligible fashion and written in standard English?

Reviewer #2: Yes

Reviewer #3: Yes

6. Review Comments to the Author

Reviewer #2: (No Response)

Reviewer #3: Overall:

Thank you for the opportunity to review this manuscript. This study develops a mathematical model to explore various aspects of Maedi-Visna (MV) virus epidemiology and makes use of three serological datasets to estimate relevant epidemiological parameters for the model. The authors then make specific management suggestions for MV control based on their findings.

The paper provides sufficient context and is easy to follow. It is written in a more conversational style and some editing could eliminate unnecessary words and hone the message, formalize some parts, and improve flow and understanding. The Results section is the most difficult to follow (but the Figures are not difficult), followed by Methods; hence some clarifications in both Methods and Results and restructuring of the text of results would greatly improve the readability of the paper for the reader. The Introduction and Discussion are full of interesting and relevant information and easier to follow and supplemental has relevant and necessary information. After addressing specific revisions suggested below, the manuscript should be ready for publication.

I agree with other Reviewers that clarity in the model comparison must be improved prior to publication and the comments on the use of the term hypothesis testing which has a specific meaning from statistics that I am not sure is intended here. I was expecting statistical hypothesis testing to appear. If an economic analysis with cost benefit ratio or similar (or a least an estimate of annual losses at some appropriate geographic/geopolitical/global unit) is available it could easily be added in a phrase or sentence without expanding too much.

Specific suggestions:

Abstract:

• Background, methods, results well included but could be modified to close with some conclusions and future directions. Many results are just listed and might be helpful to put in context. Ending could be made up of the last sentence (or two) of the Introduction, which sound more like they belong somewhere in the middle and end of the abstract.

• The latency sentence does not clearly describe your finding due to the punctuation. Instead of colon, consider putting the definition between commas or in parenthesis from “a time… …infectiousness”.

• What do you think the mechanism behind poor body condition being associated with more rapid seroconversion is? Interesting finding. You explain this in the Discussion but you may be able to hone abstract enough to discuss a few of the results points briefly here.

• Replacement lambs – what exactly does this term mean? From the abstract it’s not clear why there is a low probability of them being infected but from the paper this is a clearly stated result. Make this sentence sound more like a result here in the abstract. “Maternal transmission was found to have a small role in transmission, as replacement lambs from infected ewes had a low probability of being infected directly by their mothers.” The slight changes make this sound more like a finding from your study not general knowledge.

Introduction:

• Lines 3-5 – citation?

• Line 9 – what other species are often infected with SRLVs?

• Line 13-14 – best split into two sentences

• Line 17 – do or do not? Unclear

• Line 21 –on average, how long is seroconversion delayed?

• Suggestion: Hone and polish this section a little further by removing unnecessary words.

• Comment: Last two paragraphs are a little choppy due to shorter sentences and a lot of information.

• Last few sentences sound like they belong in the middle (copied ‘We applied our…) and end of the abstract (moved not copied ‘Our results provide…’) instead of here at the end of the Introduction.

Methods:

• “and some other ewes were removed.” – unclear

• Data set descriptions: Please include the manufacturer of the indirect ELISA kit, what is the sensitivity and specificity of the kit for MV virus in each of the species tested.

• Model description:

o What would be the proposed mechanism of resistance to MV infection? Resistance and what is known about the mechanisms should be described in the Introduction as this seems to be the first time it is considered in detail. The answer to this question appears to be done later in lines 206-210. Perhaps this should be earlier.

o Assumptions section – first four bullets are clearly assumptions. Subsequent bullets mix assumptions and what was tested. Perhaps keep this section assumptions and then have what was tested in another section of bullets? This is a really long block of bullets and not all text is assumptions.

o

o Is this a reasonable assumption? Do you think the empirical data suggest variability in ewe infectiousness? Was sensitivity analysis conducted of a few example situations in which ewes are not equally infectious and in which infectivity does not remain constant (use various patterns over time) to see the impact on model dynamics? Anything from the literature on this?

o Did you test transmission scaling intermediate between the extremes of density dependence and frequency dependence? This has been done in other systems such as in Smith et al 2009 PNAS (Host-pathogen time series data in wildlife support a transmission function between density and frequency dependence) and Cross 2013 Ecology (‘Female Elk contacts are neither frequency nor density dependent’) If you did not and no plans to, maybe discuss.

o What is the impact of your assumption of varying which months were on pasture in the face of the lack of data on duration of housing each month?

o For latent and seroconversion period, why were those distributions (exponential and gamma) assumed and why were the specific distributions tested? This is mentioned a bit in 156-159 and might be nice to have these closer together.

o Bullet at bottom of page 4, lines 138 – 142. This is unclear and although it contains and assumption and a test starts to be part of something that perhaps should be prose?

o Line 147: So there is little to no maternal immunity? What is known on maternal immunity should be briefly included in Introduction. So far I only see in the Abstract and here. Later on I find in the caption to Figure 7 with a Reference. And then a nice section in Discussion. Having a hint more mention in the beginning would be helpful.

o Line 167 – should this be ‘up to three’?

o Line 194 – spell out AIC first time and give acronym for SSR here ahead of it’s usage a few lines later

o Please explain how you will definite a significant enough difference in SSR or AIC to differentiate between models.

o Table 1 – why is field transmission exponential but household transmission rate gamma distributed? Suggest explaining in text.

o What language and/or programs was the model coded in and what version?

o Unclear how determining resistant ewes will be used later in this analysis or for providing inference for control.

o Line 220: How many age classes in model?

o Lines 226-232: suggest adding some results (in form of figure, etc) to the supplementary information so reader can make comparison. It is a bit unclear what occurred in lines 226 to the end of the section so hard to assess appropriateness of methods.

o Line 234: cite appropriate references for Next Generation Matrix (example van den Driessche etc)

o Overall, methods could benefit from a little more information on the mathematical model that is instead located in the supplement.

Results:

• This is the most challenging part of the paper for the reader, as it is hard to follow and to know what is important finding and interesting finding. Some of the text appears to be a Discussion of the results. Figures are fine (but wish they were located by their captions).

• Houwers 2. Donor infectiousness – need to explain what a difference of around 3 in AIC means for decision making. Is this strong evidence or not? While the smallest AIC was usually selected for several comparisons, the difference in AIC was not large and what each difference means for how different each model is should be better explained in Methods and Results. Also explain why you make the conclusion in line 281.

• Overall, am not sure if the test in 2 and 3 are needed for this paper. It seems the main findings and take aways are about the distribution of seroconversion time, impact of housed vs pasture management, etc. Yet the Methods and Results keep being listed in order. Perhaps some restructuring of main findings vs additional findings would help flow for reader. Table 2 does help make it easier to follow results in the Houwers model section.

• “density-dependent transmission, in which field and housed transmission rates are independent” – this was an unexpected definition of DD transmission (can’t densities also be high on the field?). Please clearly state your definitions of the transmission scaling types in the Methods. Some of the text of this Results section belongs in the Methods so just the Results are reported in Results. Is the AIC difference between frequency and density dependence sufficiently large to say there is an important difference? Same question with section 6 Distribution shape. Are the differences large enough to conclude something?

• “Given the assumption that seroconversion period is Gamma distributed, this means that it may take roughly 1.5 years for 95% of healthy ewes to seroconvert and 2.5 months for 95% of ewes with poor 318 body condition.” This calculation is unclear, please explain further. Lines 316-318.

• Lines 364-365:” months for R0 values in typical commercial flocks that house tens of ewes together over winter and for lambing” – citation? What is the range of those R0s? Can they be highlighted on Figure 8? Also some of the paragraph seems like explanation that belongs in Discussion.

• What are the main findings from the Results? At the end I am a bit confused, suggesting restructuring could help the reader follow the arguments and focus on important findings, their meanings, and implications for control. Takeaways become much clearer in Discussion.

Discussion:

• Overall: thoughtful with sufficient references and really interesting

• Latencies – some of this information sounds more like it belongs in Introduction.

• Line 302: cv?

• Line 472: based on the findings before it – is this further research needed (sounds mostly like things that were covered already by previous research)? If not, what else is missing that you would recommend?

• Line 518: Perhaps including the R0 equation for this model system earlier in the paper might be helpful. (also see comment below on supplemental information).

Conclusions:

• Line 547 – synthesize?

• Line 556 – Suggest rewording last phrase

• Does second paragraph belong in Discussion and not in the Conclusions?

Supplemental:

• Pg 1 equation 2: what is the o in o(delta t)?

• Where is the derivation of the R0 (or was it simply only obtained from next gen matrix)?

• Add references for next gen matrix to section 2 or reference list.

7. PLOS authors have the option to publish the peer review history of their article (what does this mean?). If published, this will include your full peer review and any attached files.

Reviewer #2: No

Reviewer #3: No

---

## [Author Response · Author response to Decision Letter 1]

10 Aug 2020

See attached file Response to reviewers

---

## [Editor Report · Decision Letter 2]

25 Aug 2020

Epidemiology and control of maedi-visna virus: curing the flock

PONE-D-19-32266R2

Dear Dr. Savill,

We’re pleased to inform you that your manuscript has been judged scientifically suitable for publication and will be formally accepted for publication once it meets all outstanding technical requirements.

Kind regards,

Stephen Raverty

Academic Editor

PLOS ONE
---

## [Editor Report · Acceptance letter]

27 Aug 2020

PONE-D-19-32266R2 

Epidemiology and control of maedi-visna virus: curing the flock 

Dear Dr. Savill:

I'm pleased to inform you that your manuscript has been deemed suitable for publication in PLOS ONE. Congratulations! Your manuscript is now with our production department. 

Kind regards, 

on behalf of

Dr. Stephen Raverty 

Academic Editor

PLOS ONE